# Associations of *MC4R*, *LEP,* and *LEPR* Polymorphisms with Obesity-Related Parameters in Childhood and Adulthood

**DOI:** 10.3390/genes12060949

**Published:** 2021-06-21

**Authors:** Asta Raskiliene, Alina Smalinskiene, Vilma Kriaucioniene, Vaiva Lesauskaite, Janina Petkeviciene

**Affiliations:** 1Health Research Institute, Faculty of Public Health, Lithuanian University of Health Sciences, 44307 Kaunas, Lithuania; vilma.kriaucioniene@lsmuni.lt (V.K.); janina.petkeviciene@lsmuni.lt (J.P.); 2Laboratory of Molecular Cardiology, Institute of Cardiology, Lithuanian University of Health Sciences, 50103 Kaunas, Lithuania; alina.smalinskiene@lsmuni.lt (A.S.); vaiva.lesauskaite@lsmuni.lt (V.L.)

**Keywords:** anthropometric measurements, single-nucleotide polymorphism, leptin-melanocortin regulation pathway, physical activity, childhood, adulthood

## Abstract

*MC4R*, *LEP*, and *LEPR* genes are involved in the hypothalamic leptin-melanocortin regulation pathway, which is important for energy homeostasis. Our study aimed to evaluate the associations between the *MC4R* rs17782313, *LEP* rs7799039, and *LEPR* rs1137101 polymorphisms with obesity-related parameters in childhood and adulthood. The data were obtained from the Kaunas Cardiovascular Risk Cohort study, which started in 1977 with 1082 participants aged 12–13 years. In 2012–2014, the follow-up survey was carried out. Genotype analysis of all respondents (*n* = 509) aged 48–49 years was performed for the gene polymorphisms using Real-Time Polymerase Chain Reaction. Anthropometric measurements were performed in childhood and adulthood. In childhood, only skinfold thicknesses were associated with gene variants being the lowest in children with *MC4R* TT genotype and *LEP* AG genotype. In adulthood, odds of obesity and metabolic syndrome was higher in *MC4R* CT/CC genotype than TT genotype carriers (OR 1.8; 95% CI 1.2–2.8 and OR 1.6; 95% CI 1.1–2.4, respectively). In men, physical activity attenuated the effect of the *MC4R* rs17782313 on obesity. The *LEP* GG genotype was associated with higher BMI, waist circumference, and visceral fat level only in men. No associations of the *LEPR* rs1137101 polymorphisms with anthropometric measurements and leptin level were found. In conclusion, the associations of the *MC4R* and *LEP* gene polymorphisms with obesity-related parameters strengthened with age.

## 1. Introduction

The high prevalence of overweight and obesity is one of the most serious public health problems of the 21st century. During the last decades, the body mass index (BMI) of children and adolescents increased globally [1]. The data of the international Health Behavior in School-Aged Children study, conducted since the 1990s, revealed the increasing prevalence of overweight, especially in boys aged 11, 13, and 15 years [2]. According to the data of another international study, WHO European Childhood Obesity Surveillance Initiative (COSI), during 10 years, a rising trend of overweight and obesity among 6–9-year-old children was observed in some participating countries, including Lithuania [3]. Evidence suggests that overweight children and adolescents are at high risk of becoming overweight adults [4]. Obesity in adulthood is associated with many comorbidities, such as cardiovascular diseases, type 2 diabetes, and cancers [5,6].

Genetic, metabolic, behavioral, and environmental factors play a role in the development of obesity [7]. Changes in body weight are related to energy balance, which is maintained via a homeostatic system. The central melanocortin system is very important for the regulation of energy homeostasis and controlling body weight [8]. Leptin (LEP), leptin receptors (LEPR), and melanocortin receptors (MC4R) are involved in the hypothalamic leptin-melanocortin regulation pathway. LEP is a hormone secreted by adipocytes. It binds to leptin receptors (LEPR) expressed on neurons in the hypothalamus and stimulates post-translational cleavage of pro-opiomelanocortin (POMC) to α- and β-melanocyte-stimulating hormone, and further signaling through MC4R [8,9]. In healthy individuals, a high LEP level leads to increased MC4R activity, a feeling of satiety, and a decreased food intake [9]. Genetic studies demonstrated that single nucleotide polymorphisms (SNPs) in the *MC4R*, *LEP*, and *LEPR* genes could be linked with obesity [10,11,12].

The *MC4R* gene has been extensively studied, during the last decades [13]. The *MC4R* rs17782313 variant was associated with overeating behavior, increased total energy and fat intake and higher BMI in childhood and adulthood [14,15,16]. The associations between the polymorphisms in *LEP* and *LEPR* genes and human obesity are still controversial. The most widely studied is the *LEP* rs7799039 polymorphism. Some studies indicated that the G allele of the *LEP* rs7799039 was associated with higher anthropometric measurements and increased risk of obesity [17,18]. On the contrary, other studies found that the A allele was linked with higher anthropometric parameters [19,20]. The rs1137101 polymorphism of the *LEPR* gene is one of the most common polymorphisms and it is believed to be associated with increased body weight and a high leptin level due to damaged capacity of *LEPR* signaling. However, only some studies proofed the associations of *LEPR* SNPs with being overweight and leptin level [21,22].

Studies of the associations between gene polymorphisms and anthropometric measurements across the life course can help to understand the mechanisms of action of genetic variants and the etiology of obesity [14,23]. The cohort study provides an opportunity to observe the study population for many years and assess the time changes in the associations between gene polymorphisms and body weight and the effect of gene-environment interaction.

Our study aimed to evaluate the associations between the *MC4R* rs17782313, *LEP* rs7799039, and *LEPR* rs1137101 polymorphisms with obesity-related parameters in childhood and adulthood.

## 2. Materials and Methods

### 2.1. Study Design and Sample

The data were obtained from the Kaunas Cardiovascular Risk Cohort study, which started in 1977 [24]. Fifteen secondary schools of Kaunas city (Lithuania) were randomly selected to participate in the study. In each selected school, all sixth-grade schoolchildren born in 1964 were examined. The data of 1082 participants aged 12–13 years were collected during the first cross-sectional survey. In 2012–2014, the fourth follow-up survey of the cohort was carried out. Over 35 years, 8.4% of participants died, 9.5% of individuals emigrated from Lithuania, and the addresses of 8.3% of individuals were not available in the National Population Register. Thus, the eligible sample consisted of 794 subjects. In total, 509 out of 794 individuals aged 48–49 years (64.1% of the eligible sample) participated in the follow-up survey. 

The study protocol was approved by the Kaunas Regional Ethics Committee for Biomedical Research (protocol number BE-2-30). Written informed consent on behalf of the children enrolled in the first survey was obtained from parents or guardians. Written informed consent for participation in the follow-up survey was obtained from all participants.

### 2.2. Measurements in Childhood

The height, weight, and thickness of subscapular and triceps skinfold were measured in childhood. The height of participants was measured to the nearest centimeter with a stadiometer. The participants wearing light indoor clothing were weighed with standard medical scales to the nearest 0.1 kg. BMI was calculated: BMI = weight(kg)/height^2^(m). Overweight and obesity were defined using age and sex specific cut-off points for BMI recommended by Cole el al. [25]. The triceps and subscapular skinfold thicknesses were measured two times to the nearest 1.0 mm with a Harpenden caliper, and the mean values were used in the analyses. The triceps skinfold was taken on the mid-line of the posterior surface of the arm (over the triceps muscle) at the level of the mid-point between the acromial and the radial. The subscapular skinfold was measured just below the angle of the scapula. The sum of the triceps and subscapular skinfold thicknesses was calculated and used in the analysis.

### 2.3. Measurements in Adulthood

In adulthood, height and weight were measured according to the same methodology as in childhood. BMI was assessed according to the WHO criteria: BMI of 25–29.9 kg/m^2^—overweight, BMI ≥ 30 kg/m^2^—obesity. Waist circumference was measured in the midline between the lower rib arch and the iliac crest using a flexible plastic measuring tape to the nearest 0.5 cm. Increased waist circumference was determined for men with waist circumference ≥ 94 cm and women ≥ 80 cm. The percentage of body fat and internal fat was assessed by bioelectrical impedance analysis using body composition analyzer OMRON (BF511). 

Blood samples for leptin, lipids (total cholesterol, low density lipoprotein (LDL) cholesterol, high density lipoprotein (HDL) cholesterol, triglycerides), and glucose measurements were taken in the morning after fasting for at least 12 h. Measurements were performed in a certified laboratory with COBAS Integra 400 plus device.

Hyperglycemia were defined at plasma glucose level ≥ 5.6 mmol/L. Participants were classified as having type 2 diabetes mellitus if they had received a diagnosis from a physician and reported the use of glucose-lowering medication. Metabolic syndrome was determined according to criteria of the International Diabetes Federation: increased waist circumference (≥94 cm for men and ≥80 cm for women) plus any two of the following: increased level of triglycerides (≥1.7 mmol/L); reduced level of HDL- cholesterol (for men < 1.03 mmol/L and for women < 1.29 mmol/L); increased level of systolic blood pressure (≥130 mm Hg) and/or diastolic blood pressure (≥85 mm Hg), or treatment of previously reported hypertension); increased fasting plasma glucose concentration (≥5.6 mmol/L) or previously diagnosed type 2 diabetes [26]. 

For dietary assessment, 24 h dietary recall method was used. Total energy intake in megajoules (MJ) per day, the percentage of energy from protein, fat, total carbohydrate, and sugars were analyzed. Physical activity was assessed using the International Physical Activity Questionnaire (IPAQ) [27]. Metabolic equivalents of task (METs) were calculated. Physical activity of less than 10 MET-hours/week was considered as low, 10–25 MET-hours/week—moderate and >25 MET-hours/week—high.

### 2.4. Genomic DNA Extraction and Genotyping

For DNA extraction, the blood samples were collected in ethylenediaminetetraacetic acid (EDTA) tubes. Genomic DNA from peripheral blood leucocytes was extracted using genomic DNA purification kit (Thermo Fisher Scientific, Waltham, MA, USA) according to the manufacturer’s recommendations. SNPs in *MC4R* gene (rs17782313) was estimated by using a commercially available genotyping kits C__32667060_10, in *LEP* gene (rs7799039)—C___1328079_10 and in *LEPR* gene (rs1137101)—C___8722581_10 (Applied Biosystems, Foster City, CA, USA). The Applied Biosystems 7900HT Real-Time Polymerase Chain Reaction System (Applied Biosystems, Foster City, CA, USA) was used for SNPs detection. The cycling program started with heating at 95 °C for 10 min, followed by 40 cycles (at 95 °C for 15 s and at 60 °C for 1 min). Finally, allelic discrimination was done by using SDS 2.3 software provided by Applied Biosystems.

### 2.5. Statistical Analysis

The normality of the distribution of continuous variables was tested by the Kolmogorov–Smirnov test. Means and standard deviations (SD) were presented for the normally distributed continuous variables, while the median and interquartile range was calculated for the distributions that did not meet the criteria of normality. Student t test and ANOVA with Bonferroni multiple comparison test were used to compare the mean values of normally distributed variables. Mann–Whitney and Kruskal–Wallis tests with Bonferroni multiple comparison tests were applied for the comparison of non-normal distributions. The categorical variables were presented as proportions, and were compared using a *χ*^2^ test and *Z*-test with Bonferroni correction for multiple comparisons. Each polymorphism was tested to ensure the fitting with Hardy–Weinberg equilibrium. Associations of gene polymorphisms with obesity, increased waist circumference, and metabolic syndrome were analyzed using multiple logistic regression analysis. The odds ratios (OR) were adjusted for sex, energy intake and physical activity level. 

Statistical data analysis was performed using the statistical package IBM SPSS Statistics for Windows, Version 20.0 (Armonk, NY, USA: IBM Corp., released 2011).

## 3. Results

Characteristics of the study participants in childhood and adulthood are presented in Table 1. In the first survey, all analyzed anthropometric measurements of girls were higher than boys; however, the prevalence of overweight and obesity did not differ by gender. Over 35 years of follow-up, BMI gain was greater in men than women. Adult men had a higher mean value of BMI and waist circumference also more visceral fat than women; however, in women, a higher body fat percentage was found. The proportion of obese men and women did not differ significantly, while a higher prevalence of overweight was observed among men than women. Serum leptin and HDL-cholesterol levels were higher in women; on the other hand, men had higher serum triglycerides and glucose level. Moreover, hyperglycemia and metabolic syndrome were more prevalent in men than women. Mean daily energy and fat intake was higher in men, while the proportion of energy from carbohydrates and sugars was higher in women. Although men spend more MET-hours/week on physical activity than women, the difference was not statistically significant.

The distributions of all analyzed genotypes did not differ from those predicted by a Hardy–Weinberg equilibrium: *p* = 0.355 for *MC4R* rs17782313, *p* = 0.697 for *LEP* rs7799039, and *p* = 0.744 for *LEPR* rs1137101. No significant differences in the frequency of *MC4R*, *LEP*, and *LEPR* genotypes or alleles between men and women were found (Table 2). The prevalence of *MC4R* rs17782313 CC genotype was very low (0.4% in men and 2.5% in women), so the two genotypes (CC and CT) were analyzed together.

In childhood, BMI was not associated with *MC4R* rs17782313, *LEP* rs7799039, and *LEPR* rs1137101 polymorphisms. *MC4R* rs17782313 C allele carriers had higher values of subscapular skinfold thickness than subjects with TT genotype, 8.5 (5.9) mm and 7.5 (3.4) mm, respectively. *LEP* rs7799039 AG genotype was associated with lower values of triceps skinfold thickness and the sum of both skinfolds’ thicknesses compared with GG genotype (Table 3).

During the follow-up period, greater BMI gain was observed in women with *MC4R* rs17782313 CT/CC genotype compared with TT genotype, 7.9 (5.3) kg/m^2^ and 6.2 (5.5) kg/m^2^, respectively (Table 4). The *MC4R* minor C allele was associated with higher mean values of BMI, waist circumference, and visceral fat level in men and women. Men with CT/CC genotype also had a higher body fat proportion than the TT genotype carriers. A higher prevalence of central obesity was found in men with CT/CC genotype compared with TT genotype, 68.4% and 51.6%, respectively. More women CT/CC genotype carriers were obese than TT genotype carriers, 29.2% and 17.4%, respectively. Leptin level did not differ by *MC4R* genotypes. The prevalence of hyperglycemia and diabetes was higher in women with CT/CC genotype compared with the TT genotype. No such association was found in men. Higher fat intake was reported by men with CT/CC genotype.

The data of multiple logistic regression analysis showed that odds of obesity, increased waist circumference, and metabolic syndrome adjusted for gender, energy intake and level of physical activity was by 1.8, 1.7, and 1.6 times higher in *MC4R* rs17782313 C allele carriers than in subjects with TT genotype (Table 5). 

The analysis of the associations of *MC4R* rs17782313 genotypes with anthropometric parameters in subgroups stratified by the level of physical activity showed that in men having a low level of physical activity, BMI, waist circumference, body fat percentage, and visceral fat level were higher in C allele carriers than in subjects with TT genotype (Table 6). In men with moderate and high-level physical activity, only the distribution of waist circumference differed between TT and CT/CC genotypes. The data of multiple logistic regression analysis showed the higher odds of obesity and increased waist circumference in men with CT/CC genotypes and a low level of physical activity. No associations with *MC4R* gene polymorphisms and anthropometric measurements were found in men with a moderate or high level of physical activity. Physical activity did not have any effect on the association between *MC4R* genotypes and anthropometric parameters in women.

Men with *LEP* rs7799039 genotype GG had higher BMI, waist circumference and visceral fat level than A allele carriers (Table 7). Moreover, men with GG genotype had the highest prevalence of obesity: 37.3% compared to 16.8% in men with AG genotype. *LEP* rs7799039 polymorphisms were not associated with anthropometric measurements in women. 

No associations of the *LEPR* rs1137101 polymorphisms with anthropometric measurements and leptin level were found (Table 8). Dietary assessment showed that women with the GG genotype reported higher fat intake than with the AA genotype.

Lipids and glucose levels did not differ between individuals with *MC4R, LEP,* and *LEPR* genetic variants. Moreover, no association between genetic variants and energy intake was found.

## 4. Discussion

This longitudinal study provides data obtained in a cohort examined in childhood (12–13 years of age) and 35 years later in adulthood (47–48 years of age). The data showed that the associations of *MC4R* rs17782313 and *LEP* rs7799039 polymorphisms with anthropometric measurements varied with age. In childhood, genetic variants had no effect on BMI; however, *MC4R* rs17782313 and *LEP* rs7799039 variants were associated with subscapular or triceps skinfold thickness. Such a weak link between genetic variants and children weight status could be partially explained by the relatively low prevalence of overweight and obesity (less than 15%). The associations of both genes’ polymorphisms with obesity-related parameters strengthened in adulthood. *MC4R* rs17782313 variants were associated with BMI, waist circumference, and visceral fat level in men and women, while *LEP* rs7799039 variants were linked with anthropometric measurements only in men. *LEPR* rs1137101 polymorphism was not associated with obesity-related parameters in childhood and adulthood.

Studies that examined the associations between gene variants and anthropometric measurements over the life course are particularly limited [14,23,28]. A study carried out in the United Kingdom demonstrated strengthening the association of *MC4R* genetic variants with BMI and weight during childhood up to age 20 years and then weakening with increasing adult age [14]. The evidence from several U.S. cohorts suggested that the effect of polygenic scores on BMI increases significantly with teenagers’ transition into adulthood [23]. In another study, *MC4R* gene polymorphism was associated with BMI change from adolescent to young adult [28]. Our study revealed greater BMI gain from childhood to adulthood in women with *MC4R* rs17782313 C allele.

We found the strongest association of *MC4R* rs17782313 variants with anthropometric parameters in adulthood. Odds of obesity, increased waist circumference, and metabolic syndrome adjusted for energy intake and level of physical activity was higher in subjects with CT/CC genotype than TT genotype. Evidence suggests that this gene is involved in appetite regulation and energy expenditure [29]. The *MC4R* rs17782313 variant was associated with the obesity-related proteins ghrelin and visfatin in the Arabic population [30]. Previous studies showed strong associations of the *MC4R* rs17782313 variant with higher energy and fat intake and lower carbohydrate and protein intake [31,32]. The dietary assessment carried out in our study estimated that the proportion of energy from fat was higher in men with the CT/CC genotype than TT genotype. 

Our findings are in line with the previous studies showing that physical activity can attenuate the effect of the *MC4R* rs17782313 on obesity [33,34,35]. Xi B et al. found significant associations between rs17782313 minor C allele and increased risk of obesity only in children with sedentary behavior [33]. The risk of obesity was higher among sedentary Spain adults with *MC4R* risky C allele [34]. The data of the HUNT study found that the genetic effect among physically inactive individuals was larger if compared to active ones [35].

In our study, the rs7799039 polymorphism in the *LEP* gene was associated with anthropometric parameters only in men. The highest values of BMI, waist circumference, and visceral fat level were observed in GG genotype carriers. Some other studies also found that G is a risk allele [17,18]. However, findings from several studies revealed that the A allele is associated with a higher risk of obesity [19,20,36]. Acute exercise decreased leptin levels in G allele carriers but increased in AA homozygotes [36]. Some authors did not report any association between *LEP* rs7799039 genetic variants and obesity-related variables [37].

The data on associations between *LEPR* rs1137101 polymorphism is also contradictory. We did not find any association between this polymorphism and body weight status. The same findings were provided by other researchers [12,38,39]. However, a significant association between *LEPR* rs1137101 minor allele and the greater weight growth was found in the Spanish children population [22]. The findings from the study carried out in Tunis also confirmed that the *LEPR* rs1137101 polymorphism influences plasma leptin levels and BMI in obese patients [21]. The *LEPR* rs1137101 G allele was associated with higher BMI and waist circumference in overweight and obese subjects in Sri Lanka [40]. The authors of the systematic review concluded that most studies were underpowered to detect small effect sizes of the polymorphism [41].

Our study did not show any association between analyzed gene polymorphism and leptin level or metabolic traits (lipids and glucose level), except higher prevalence of hyperglycemia and diabetes in women with CT/CC genotype. This association can be explained by higher obesity prevalence in this group.

The strength of the present study is the randomly selected cohort examined in childhood and after 35 years. All anthropometric measurements were taken using the same standardized methodology in both surveys. A further strength includes the adjustment for energy intake and physical activity level in logistic regression analysis for the associations of gene polymorphisms with obesity and metabolic syndrome. The study has several limitations. During the 35-year follow-up, the loss of participants was quite substantial. A high rate of emigration from Lithuania over the last decades was one of the causes for non-participation in the follow-up survey. However, our previous study did not identify the differences in baseline measurements between participants and non-participants [42]. Finally, we did not have data on lipid and glucose levels, or nutrition and physical activity habits in the baseline survey, and we could not analyze the associations between gene variants and those variables. 

## 5. Conclusions

The identified associations of *MC4R* and *LEP* polymorphisms with obesity-related parameters should be considered as risk factors for age- and gender-related obesity development. The *LEPR* polymorphism was not associated with anthropometric measurements. Physical activity attenuates the effect of the *MC4R* polymorphism on obesity in men. Further lifestyle intervention studies are needed to assess the effect of physical activity on body weight in obese individuals with different genes polymorphisms.

## Figures and Tables

**Table 1 genes-12-00949-t001:** Characteristics of the study sample.

Characteristics	Boys/Men *n* = 230	Girls/Women *n* = 279	*p*-Value
Childhood (12–13 years); mean (SD) or median (IQR)			
Age (years) *	12.9 (0.4)	12.9 (0.6)	0.979
Weight (kg) **	42.9 (13.1)	47.1 (13.0)	<0.001
BMI (kg/m^2^) **	17.8 (3.4)	18.6 (3.3)	0.003
Skinfold thickness (mm):			
subscapular **	5.5 (2.4)	7.3 (3.8)	<0.001
Triceps **	10.2 (5.1)	13.7 (6.4)	<0.001
sum of both **	15.6 (7.0)	21.2 (9.3)	<0.001
Childhood (12–13 years); percentages			
Overweight	11.3	12.7	0.623
Obesity	1.7	2.5	0.528
Adulthood (48–49 years); mean (SD) or median (IQR)			
Age (years) *	48.0 (0.2)	48.1 (0.2)	0.045
Increase in BMI (child–adult) (kg/m^2^) **	8.8 (5.6)	6.8 (5.7)	<0.001
BMI (kg/m^2^) **	26.9 (5.5)	25.6 (6.3)	0.009
Waist circumference (cm) **	95.8 (16.3)	82.0 (16.3)	<0.001
Body fat percentage **	25.0 (9.0)	38.0 (10.0)	<0.001
Visceral fat level **	11.0 (7.0)	7.0 (3.0)	<0.001
Leptin (ng/mL) **	4.3 (4.8)	12.1 (12.1)	<0.001
Cholesterol (mmol/L) *	6.1 (1.1)	6.2 (1.1)	0.543
LDL-cholesterol (mmol/L) *	3.9 (1.0)	3.8 (1.1)	0.105
HDL-cholesterol (mmol/L) *	1.4 (0.5)	1.8 (0.6)	<0.001
Triglycerides (mmol/L) **	1.3 (0.8)	1.0 (0.5)	<0.001
Glucose (mmol/L) **	5.3 (0.8)	5.0 (0.6)	<0.001
Energy intake (MJ/d) **	8.8 (4.3)	6.3 (3.4)	<0.001
Protein intake (E%) *	14.1 (5.5)	13.8 (6.0)	0.931
Fat intake (E%) *	40.7 (12.5)	37.7 (13.3)	0.009
Carbohydrate intake (E%) *	43.0 (15.0)	45.9 (15.4)	<0.001
Sugar intake (E%) *	13.8 (8.9)	16.7 (10.7)	<0.001
Physical activity * (MET-hours/week)	50.1 (126.4)	39.1 (105.7)	0.411
Adulthood (48–49 years); percentages			
Overweight	43.9	34.1	0.043
Obesity	25.2	21.1	0.275
Increased waist circumference	57.0	57.3	0.453
Hyperglycemia	29.6	13.3	<0.001
Diabetes	3.9	2.2	0.242
Metabolic syndrome	33.6	18.3	<0.001
Physical activity levels:			0.740
low	25.2	26.2	
moderate	31.3	33.7	
high	43.5	40.1	

* Mean and standard deviation; ** median and interquartile range; SD—standard deviation; IQR—interquartile range; BMI—body mass index; LDL—low-density lipoprotein; HDL—high-density lipoprotein; MJ/d—megajoules per day; E%—energy percentage; MET—metabolic equivalent of task.

**Table 2 genes-12-00949-t002:** Distribution of genotypes and allele frequencies (%).

Genotype	Men	Women	*p*-Value
	*n* = 230	*n* = 279	
*MC4R* rs17782313			
TT	66.8	68.1	
CT	32.8	29.4	0.141
CC	0.4	2.5	
T allele	0.83	0.83	0.866
C allele	0.17	0.17	
*LEP* rs7799039			
GG	29.3	36.4	
AG	54.6	47.1	0.19
AA	16.1	16.5	
G allele	0.57	0.6	0.334
A allele	0.43	0.4	
*LEPR* rs1137101			
AA	21.8	23.9	
AG	50.5	52.5	0.555
GG	27.7	23.6	
A allele	0.47	0.5	0.34
G allele	0.53	0.5	

**Table 3 genes-12-00949-t003:** Median (IQR) of anthropometric measurements by *MC4R*, *LEP,* and *LEPR* genotypes in childhood.

Genotype	Body Mass Index (kg/m^2^)	Subscapular Skinfold Thickness (mm)	Triceps Skinfold Thickness (mm)	Sum of Skinfold Thicknesses (mm)
*MC4R* rs17782313				
TT	18.3 (3.3)	7.5 (3.4)	11.9 (6.1)	18.5 (9.3)
CT/CC	18.6 (3.6)	8.5 (5.9)	12.3 (7.8)	18.8 (10.6)
*p*-Value (χ^2^ test)	0.123	0.009	0.399	0.309
*LEP* rs7799039				
GG	18.4 (3.8)	8.3 (4.7)	12.5 (7.5)	19.2 (12.2)
AG	18.2 (3.2)	7.5 (4.4) *	11.2 (6.8) *	17.5 (9.5) *
AA	18.4 (3.5)	7.6 (3.0)	12.2 (5.7)	19.6 (9.4)
*p*-Value (χ^2^ test)	0.288	0.192	0.042 *	0.031 *
*LEPR* rs1137101				
AA	18.3 (3.4)	8.1 (4.6)	12.1 (7.9)	19.1 (12.1)
AG	18.3 (3.3)	7.9 (4.7)	11.9 (6.9)	18.4 (10.2)
GG	18.4 (3.3)	7.6 (36)	12.2 (6.7)	18.5 (9.7)
*p*-Value (χ^2^ test)	0.842	0.769	0.386	0.595

* *p* < 0.05 compared *LEP* rs7799039 genotype AG with GG (Kruskal–Wallis test with Bonferroni correction for multiple comparisons); SD—standard deviation; IQR—interquartile range.

**Table 4 genes-12-00949-t004:** Association of the *MC4R* rs17782313 polymorphism with anthropometric measurements and other factors in adulthood.

Characteristics	Men			Women		
	TT	CT/CC	*p*-Value	TT	CT/CC	*p*-Value
	Mean (SD) or Median (IQR)			Mean (SD) or Median (IQR)		
Increase in BMI (child–adult) (kg/m^2^) **	8.4 (5.7)	9.2 (5.0)	0.085	6.2 (5.5)	7.9 (5.3)	0.01
BMI (kg/m^2^) **	26.1 (5.9)	28.2 (6.7)	0.006	25.4 (6.0)	26.4 (9.0)	0.025
Waist circumference (cm) **	95.0 (17.2)	99.0 (17.8)	0.003	81.1 (15.1)	85.0 (17.3)	0.033
Body fat percentage **	25.0 (9.0)	27.0 (10.0)	0.033	37.5 (11.0)	38.0 (10.0)	0.199
Visceral fat level **	10.0 (6.0)	13.0 (7.0)	0.007	7.0 (3.0)	8.0 (3.0)	0.037
Leptin (ng/mL) **	4.2 (4.4)	4.2 (4.8)	0.100	11.6 (10.9)	12.9 (12.9)	0.208
Fat intake (E%) *	39.2 (12.0)	42.7 (13.1)	0.025	37.5 (13.2)	38.5 (13.9)	0.754
	Percentages			Percentages		
Obesity	22.2	31.6	0.125	17.4	29.2	0.024
Increased waist circumference	51.6	68.4	0.016	54.2	64.0	0.122
Hyperglycemia	30.7	26.3	0.490	10.5	19.1	0.049
Diabetes	3.9	3.9	0.992	0.5	5.6	0.006
Metabolic syndrome	30.3	40.6	0.113	15.8	23.9	0.106

* Mean and standard deviation; ** median and interquartile range.

**Table 5 genes-12-00949-t005:** Odds ratios for obesity, waist circumference, and metabolic syndrome by *MC4R* rs17782313 genotypes in adulthood.

Variable	*MC4R* rs17782313 Genotype			
	TT	CT/CC		
		Odds Ratio *	95% Confidence Intervals	*p*-Value
Obesity	1	1.8	1.2–2.8	0.006
Increased waist circumference	1	1.7	1.2–2.6	0.005
Metabolic syndrome	1	1.6	1.1–2.4	0.032

* Odds ratios adjusted for gender, energy intake, and level of physical activity.

**Table 6 genes-12-00949-t006:** Association of the *MC4R* rs17782313 polymorphism with anthropometric measurements by the level of physical activity in men.

Characteristics	Low Level of Physical Activity			Moderate and High Level of Physical Activity		
	Median (IQR)			Median (IQR))		
	TT	CT/CC	*p*-Value	TT	CT/CC	*p*-Value
BMI (kg/m^2^)	25.8 (5.7)	30.3 (5.4)	0.007	26.2 (6.0)	27.6 (9.2)	0.088
Waist circumference (cm)	97.0 (14.7)	103.0 (13.4)	0.023	93.2 (17.5)	97.2 (16.3)	0.030
Body fat percentage	26.0 (9.0)	28.5 (9.0)	0.035	24.0 (9.0)	26.0 (10.0)	0.197
Visceral fat level	10.0 (7.0)	15.0 (5.0)	0.010	11.0 (7.0)	12.0 (6.0)	0.062
	Odds ratio * (95% CI)			Odds ratio * (95% CI)		
Obesity	1	5.6 (1.6–19.8)	0.007	1	1.0 (0.5–2.2)	0.850
Increased waist circumference	1	4.2 (1.0–17.0)	0.048	1	1.7 (0.9–3.3)	0.123

* Odds ratios adjusted for gender, energy intake and level of physical activity; IQR—interquartile range; CI—confidence interval.

**Table 7 genes-12-00949-t007:** Association of the *LEP* rs7799039 polymorphism with anthropometric measurements and leptin level in adulthood.

Characteristics	Men				Women			
	GG	AG	AA	*p*-Value	GG	AG	AA	*p*-Value
	Median (IQR)				Median (IQR)			
Increase in BMI (child–adult) (kg/m^2^)	10.2 (7.1)	8.5 (5.0)	9.0 (5.7)	0.114	6.7 (5.6)	6.5 (5.0)	8.1 (6.9)	0.607
BMI (kg/m^2^)	27.9 (8.8)	26.7 (4.3) *	26.0 (8.8) *	0.040	25.6 (6.1)	25.3 (6.1)	27.1 (7.7)	0.354
Waist circumference (cm)	100.4 (14.1)	95.6 (11.0) *	96.2 (13.6)	0.035	81.0 (15.9)	82.0 (15.5)	85.0 (17.5)	0.768
Visceral fat level	12.0 (8.0)	11.0 (5.0 *)	10.5 (9.0) *	0.037	7.0 (3.0)	7.0 (3.0)	8.0 (4.0)	0.352
Leptin (ng/mL)	5.1 (5.6)	3.9 (3.9)	3.8 (4.1)	0.251	11.2 (10.9)	12.3 (13.4)	13.0 (9.8)	0.982
	Percentages				Percentages			
Obesity	37.3	16.8 **	29.7	0.006	16.2	20.3	28.9	0.211
Increased waist circumference	65.7	52.0	56.8	0.190	55.6	56.3	64.4	0.571

* *p* < 0.05 compared with *LEP* rs7799039 GG genotype (Kruskal–Wallis test with Bonferroni correction for multiple comparisons); ** *p* < 0.05 compared with *LEP* rs7799039 GG genotype (Z-test with Bonferroni correction for multiple comparisons); SD—standard deviation; IQR—interquartile range.

**Table 8 genes-12-00949-t008:** Association of the *LEPR* rs1137101 polymorphisms with anthropometric measurements and other factors in adulthood.

Characteristics	Men				Women			
	AA	AG	GG	*p*-Value	AA	AG	GG	*p*-Value
	Mean (SD) or Median (IQR)				Mean (SD) or Median (IQR)			
Increase in BMI (child–adult) (kg/m^2^) **	8.0 (5.3)	9.8 (5.9)	7.9 (5.4)	0.214	6.6 (6.0)	6.6 (5.0)	7.8 (6.6)	0.655
BMI (kg/m^2^) **	25.8 (5.9)	27.7 (5.5)	26.3 (6.4)	0.201	25.4 (8.0)	25.4 (6.2)	26.2 (6.9)	0.422
Waist circumference (cm) **	95.0 (16.3)	98.2 (14.8)	95.0 (16.3)	0.168	84.0 (20.5)	81.2 (16.3)	84.3 (15.3)	0.283
Visceral fat level **	10.0 (8.0)	12.0 (6.0)	11.0 (8.0)	0.176	7.0 (4.0)	7.0 (3.0)	8.0 (3.0)	0.421
Leptin (ng/mL) **	4.2 (4.0)	4.9 (4.9)	4.3 (3.5)	0.662	10.4 (11.8)	11.0 (11.7)	15.1 (13.5)	0.348
Fat intake (E%) *	42.1 (15.4)	41.2 (11.6)	39.5 (13.0)	0.749	34.8 (11.0)	38.3 (13.5)	41.5 (14.3) *	0.015
	Percentages				Percentages			
Obesity	22.9	27.0	26.2	0.861	27.3	17.9	23.1	0.286
Increased waist circumference	52.1	64.0	52.5	0.214	57.6	54.5	64.6	0.389

* *p* < 0.05 compared with *LEPR* rs1137101 AA genotype (Kruskal–Wallis test with Bonferroni correction for multiple comparisons); SD—standard deviation; IQR—interquartile range; * mean and standard deviation; ** median and interquartile range.

## Data Availability

The data presented in this study are available upon request from the corresponding author. The data are not publicly available due to ethical issues.

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
