# Peer review of "Associations of MC4R, LEP, and LEPR Polymorphisms with Obesity-Related Parameters in Childhood and Adulthood"

_genes, 2021, doi:10.3390/genes12060949_

Round 1
Reviewer 1 Report
Please see the attached review.

Reviewer 2 Report
I would like to thank the Authors and Editors for the opportunity to review this paper.
I find the manuscript very interesting, and the Authors have put a lot of effort in their analytic methods and calculations. Methodology seems appropriate, the results are clearly described, the discussions are properly supported by the results.
The concluding chapter is a little bit quick, I would just suggest to flesh it out a little bit more, otherwise it appears as an afterthought.
One question that I thought about was the following: would you deem it possible or useful to collect the data from the second or third follow-up surveys, as middle points through the data that you have used?
I would advise to re-read the text, with particular attention to the text in the Tables, which is sometimes incorrect (missing patrentheses, minor spelling mistakes).
Congratulations for the good work, for me the manuscript has to be just slightly refined.
Round 2
Reviewer 1 Report
Please see the attached review.
